

# Drinking water and rural schools in the Western Amazon: an environmental intervention study

Maura Regina Ribeiro[1], Luiz Carlos de Abreu[1,2] and Gabriel Zorello Laporta[1,3]

[1] Setor de Pós-graduação, Pesquisa e Inovação, Faculdade de Medicina do ABC, Santo André, São Paulo, Brazil
[2] Programa de Pós-graduação em Políticas Públicas e Desenvolvimento Local, EMESCAM, Vitória, Espírito Santo, Brazil
[3] Centro de Engenharia, Modelagem e Ciências Sociais Aplicadas, Universidade Federal do ABC, Santo André, São Paulo, Brazil

Corresponding author
Gabriel Zorello Laporta,
gabriel.laporta@fmabc.br

## ABSTRACT

**Background:** Although water and sanitation are considered human rights, worldwide approximately three of 10 people (2.1 billion) do not have access to safe drinking water. In 2016, 5.6 million students were enrolled in the 34% of Brazilian schools located in rural areas, but only 72% had a public water supply network. The objective was to evaluate effectiveness of environmental intervention for water treatment in rural schools of the Western Amazonia, and determine the efficacy of water treatment using a simplified chlorinator on potability standards for turbidity, fecal coliforms and *Escherichia coli.*
**Methods:** A simplified chlorinator was installed for treatment of potable water in 20 public schools in the rural area of Rio Branco municipality, Acre state, Brazil.
**Results:** Before the intervention, 20% ($n = 4$), 100% ($n = 20$) and 70% ($n = 14$) of schools had water that failed to meet potability standards for turbidity, fecal coliforms and *E. coli*, respectively. However, after intervention, 70% ($p = 0.68$), 75% ($p < 0.001$) and 100% ($p < 0.001$) of schools complied with potability standards.
**Discussion:** This intervention considerably improved schools' water quality, thus decreasing children's health vulnerability due to inadequate water. Ancillary activities including training, educational lectures, installation of equipment, supply of materials and supplies (65% calcium hypochlorite and reagents) were considered fundamental to achieving success full outcomes. Installation of a simplified chlorinator in rural schools of the Western Amazon is therefore proposed as a social technology aiming at social inclusion, as well as economic and environmental sustainability.

## INTRODUCTION

Depriving people of access to safe drinking water denies them the right to life (*Zorzi, Turatti & Mazzarino, 2016*). Although water and sanitation are considered a human right

and basic conditions for human health, dignity, economic development and social well-being (*Malhotra, Sidhu & Devi, 2015*), worldwide approximately three of 10 people (2.1 billion) do not have access to safe drinking water and six of 10 (4.5 billion) lack adequate sanitation (*United Nations Economic and Social Council (Ecosoc), 2017*). The need for access to drinking water and sanitation is a public health issue (*Jaravani et al., 2016*), since its unavailability increases incidence of infectious diseases, including diarrhea, cholera, hepatitis A and typhoid fever, among others (*United Nations Economic and Social Council (Ecosoc), 2017*).

In 2016, 5.6 million students were enrolled in the 34% of Brazilian schools located in rural areas, with only 72% having a public water supply network (*Ministério da Educação, 2016*). The Unified Health System (SUS) policy recognizes sanitation as central in improving health conditions. Notwithstanding, most of the rural populations in the Amazon region live under adverse conditions and are deprived of potable drinking water due to inadequate services, infrastructure and operational facilities. Our objective was to provide effective treatment of water for human consumption in rural schools in the city of Rio Branco (AC), Western Amazon, Brazil, and determine the efficacy of water treatment using a simplified chlorinator on potability standards for turbidity, fecal coliforms, and *Escherichia coli*.

# MATERIALS AND METHODS

## Study design

The study was an environmental intervention for treatment of water for human consumption, through installation of a simplified chlorinator, in 20 public schools in the rural area of Rio Branco, AC. The steps involved are shown (Fig. 1).

## Study area and studied population

The study was conducted in the rural area of the municipality of Rio Branco, located in the extreme southwest of the North Region, in the Brazilian Amazon (Table 1; Fig. 2). With an estimated population (in 2016) of 377,057 inhabitants, the municipality has an area of 8,836 km$^2$. The population living in rural areas was 27,493 (*IBGE, 2010*) according to the 2010 Demographic Census.

## Environmental intervention: chlorinator

The simplified chlorinator is a social technology recommended by the Brazilian Agricultural Research Corporation (EMBRAPA) of São Carlos, SP, as a domestic method of disinfecting water in rural areas by facilitating addition of chlorine into water reservoirs (boxes of water). This device was made of 25 mm welded PVC tubes, ¾ inch ball register, ¾ inch male adapter threaded PVC, 25 × ¾ inch adapter, ¼ inch garden faucet, 25 × ¼ inch tube T-form connections, 60 × 25 inch, assembled with glue and sandpaper, and installed between the water intake port and the reservoir system (Fig. 3).

## Sampling drinking water in rural schools

In each school, 300 mL water was collected before and after treatment (canopy tap), with 100 mL for microbiological analysis and 200 mL for organoleptic tests. For post-

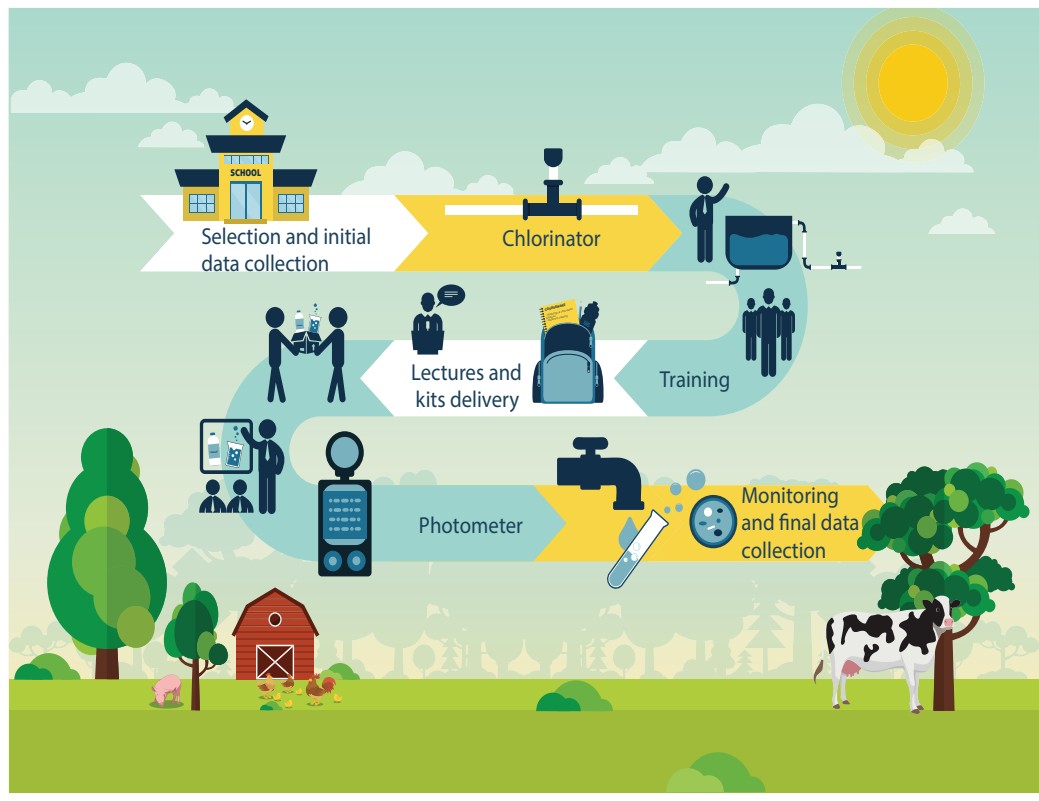

**Figure 1 Infographic of the environmental intervention study.** Strategic plan and action steps during the environmental intervention study. Intervention means both chlorinator installation and environmental education approach (training, lectures and kit delivery) in the rural schools.

intervention water samples, sterile plastic sachets containing sodium thiosulfate pellets were used for neutralization of chlorine present in the sample. Samples were then placed in a sterile box containing ice and transported to the Central Analysis Laboratory (LACEN) of the State Health Department (SESACRE, Rio Branco, Acre, Brazil).

Prior to each water collection, some procedures were adopted for the purpose of preserving samples, including cleaning taps with 70% alcohol, washing and disinfection of hands with soap and 70% alcohol, enumeration of sterile plastic bags, allowing tap water to flow for 2–3 min and filling the bag ~3/4 full (to allow thorough mixing).

For organoleptic analysis, the nephelometric method recommended by the Standard Methods for Examination of Water and Wastewater (SMEWW), 21st Ed. 2130B, expressed in nephelometric units of turbidity (uT) was adopted. For microbiological analysis, the chromogenic/enzymatic substrate method was used, also recommended by the SMEWW, 21st Ed. 9223B, using a Colilert test kit and incubating samples for bacteriological culture at an incubator (Digital model Q-316-M5) at 35 °C for 24 h.

## Water parameters

Microbiological (total coliforms and *E. coli*) and organoleptic (turbidity) parameters were assessed before and after the intervention. Maximum permitted values (MPV) established

**Table 1 Schools analyzed in the rural area of the municipality of Rio Branco, Acre state.**

| School I.D. | Management | Number of students | Age Min. | Max. | Source of supply | Intervention |
|---|---|---|---|---|---|---|
| 1 | State | 24 | 8 | 40 | Acre river | No |
| 2 | State | 10 | 6 | 13 | Amazonas–Cacimbão well | No |
| 3 | State | 65 | 4 | 50 | Amazonas–Cacimbão well/Water truck | No |
| **4** | **State** | **25** | **6** | **12** | **Amazonas–Cacimbão well** | **Yes** |
| 5 | State | 50 | 6 | 20 | Local river | No |
| 6 | State | 147 | 6 | 61 | Local water reservoir | No |
| 7 | State | 105 | 6 | 13 | Semi-artesian well | No |
| **8** | **Municipal** | **83** | **3** | **6** | **Amazonas–Cacimbão well** | **Yes** |
| 9 | State | 25 | 6 | 12 | Semi-artesian well | No |
| 10 | State | 17 | 6 | 13 | Amazonas–Cacimbão well | No |
| **11** | **State** | **70** | **6** | **55** | **Amazonas–Cacimbão well** | **Yes** |
| 12 | State | 10 | 6 | 13 | Amazonas–Cacimbão well | No |
| 13 | State | 15 | 5 | 12 | Amazonas–Cacimbão well | No |
| 14 | State | 24 | 7 | 24 | Amazonas–Cacimbão well | No |
| 15 | State | 29 | 7 | 26 | Iguarape | No |
| **16** | **State** | **129** | **7** | **20** | **Amazonas–Cacimbão well** | **Yes** |
| **17** | **State** | **145** | **6** | **20** | **Amazonas–Cacimbão well** | **Yes** |
| 18 | State | 21 | 6 | 14 | Semi-artesian well | No |
| 19 | State | 572 | 8 | 22 | Local water reservoir/Water truck | No |
| 20 | State | NI* | 6 | 17 | Semi-artesian well | No |
| **21** | **Municipal** | **210** | **4** | **15** | **Semi-artesian well** | **Yes** |
| 22 | State | 350 | 10 | 29 | Semi-artesian well | No |
| 23 | State | 101 | 5 | 28 | Semi-artesian well | No |
| 24 | State | 15 | 6 | 13 | Amazonas–Cacimbão well | No |
| 25 | State | 187 | 4 | 14 | Water truck | No |
| 26 | State | 225 | 6 | 12 | Water truck | No |
| 27 | State | 21 | 7 | 12 | Local water reservoir | No |
| 28 | State | 9 | 6 | 11 | Amazonas–Cacimbão well | No |
| **29** | **State** | **100** | **4** | **20** | **Amazonas–Cacimbão well** | **Yes** |
| 30 | State | 36 | 6 | 26 | Amazonas–Cacimbão well | No |
| 31 | State | 25 | 5 | 35 | Amazonas–Cacimbão well | No |
| 32 | State | 13 | 6 | 10 | Semi-artesian well | No |
| 33 | State | 10 | 6 | 10 | Amazonas–Cacimbão well | No |
| **34** | **State** | **298** | **6** | **21** | **Amazonas–Cacimbão well** | **Yes** |
| 35 | State | 16 | 8 | 15 | Local river | No |
| **36** | **State** | **440** | **6** | **60** | **Semi-artesian well** | **Yes** |
| 37 | State | 102 | 6 | 17 | Semi-artesian well | No |
| **38** | **Municipal** | **40** | **2** | **3** | **Semi-artesian well** | **Yes** |

| School I.D. | Management | Number of students | Age | | Source of supply | Intervention |
|---|---|---|---|---|---|---|
| | | | Min. | Max. | | |
| **39** | **Municipal** | **186** | **6** | **18** | **Amazonas–Cacimbão well/Water truck** | **Yes** |
| 40 | Municipal | 363 | 6 | 14 | Semi-artesian well | No |
| 41 | State | 38 | 6 | 13 | Amazonas–Cacimbão well | No |
| **42** | **Municipal** | **12** | **7** | **14** | **Amazonas–Cacimbão well** | **Yes** |
| 43 | State | 16 | 7 | 14 | Amazonas–Cacimbão well | No |
| **44** | **State** | **162** | **6** | **25** | **Amazonas–Cacimbão well** | **Yes** |
| **45** | **State** | **26** | **5** | **12** | **Semi-artesian well** | **Yes** |
| 46 | State | 24 | 6 | 17 | Iguarape | No |
| 47 | State | 194 | 4 | 14 | Water truck | No |
| 48 | Municipal | 50 | 6 | 18 | Amazonas–Cacimbão well | No |
| **49** | **Municipal** | **218** | **4** | **13** | **Semi-artesian well** | **Yes** |
| **50** | **Municipal** | **25** | **6** | **15** | **Amazonas–Cacimbão well** | **Yes** |
| 51 | State | 27 | 5 | 12 | Amazonas–Cacimbão well | No |
| 52 | State | 198 | 5 | 14 | Semi-artesian well | No |
| 53 | State | 310 | 6 | 22 | Semi-artesian well/Water truck | No |
| **54** | **State** | **336** | **11** | **17** | **Semi-artesian well** | **Yes** |
| 55 | State | 222 | 4 | 26 | Semi-artesian well | No |
| 56 | State | 45 | 6 | 58 | Amazonas–Cacimbão well | No |
| 57 | Municipal | 30 | 6 | 14 | Semi-artesian well | No |
| 58 | State | 7 | 6 | 14 | Local water reservoir | No |
| 59 | State | 31 | 15 | 24 | Amazonas–Cacimbão well | No |
| 60 | Municipal | 65 | 6 | 70 | Amazonas–Cacimbão well | No |
| 61 | State | 55 | 7 | 15 | Amazonas–Cacimbão well | No |
| 62 | Municipal | 23 | 6 | 13 | Amazonas–Cacimbão well | No |
| **63** | **State** | **130** | **5** | **45** | **Amazonas–Cacimbão well** | **Yes** |
| **64** | **Municipal** | **186** | **4** | **14** | **Water truck** | **Yes** |
| **65** | **State** | **125** | **6** | **12** | **Semi-artesian well** | **Yes** |

**Notes:**
Type of administration, number of students enrolled, minimum and maximum students age, source of water supply and intervention status.
The entries in bold represent selected schools that participated in this environmental intervention study (see the pink triangles in Fig. 2).

by Ministry of Health (MS) Ordinance No. 2914/2011 for *E. coli* and total coliforms are absence in 100 mL, the former being an indicator of fecal contamination and the second an indicator of treatment efficiency. For turbidity, MPV is 5.0 units of turbidity (uT).

## Water chlorination

Water was chlorinated with granulated calcium hypochlorite (65%), diluted with water in a plastic container and immediately added to the simplified chlorinator installed in each school, at a dosage of one teaspoon per 1,000 L of water. To monitor free residual chlorine, a HANNA brand portable Colorimeter Checker was distributed to each school.
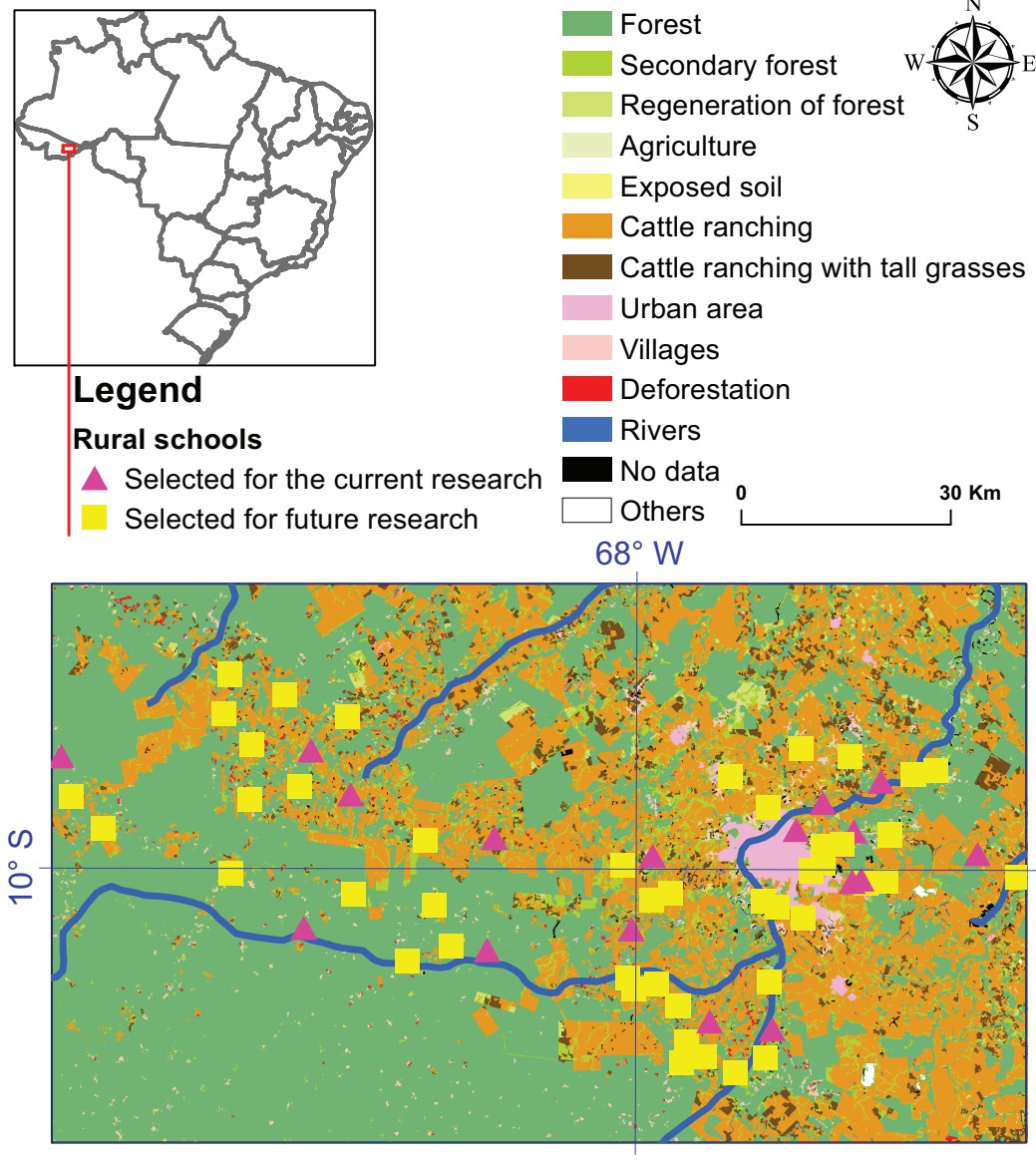

**Figure 2** **Location map of the 65 schools in the rural area of Rio Branco municipality, Acre state.** Schools with intervention (pink triangles) were studied in the present study and schools without intervention (yellow squares) were selected for a future research with a wider health care intervention approach.

This instrument measures free chlorine concentration (parts per million, ppm) and the result must be between 0.20 mg/L (minimum) and 2.0 mg/L (maximum), according to MS Ordinance No. 2914/2011.

## Data analysis

McNemar's Chi-square test was used for count data in the R programming environment v. 3.3.0, to test for symmetry of rows and columns in a two-dimensional contingency table with the following variables: (1) water potability before intervention (satisfactory,

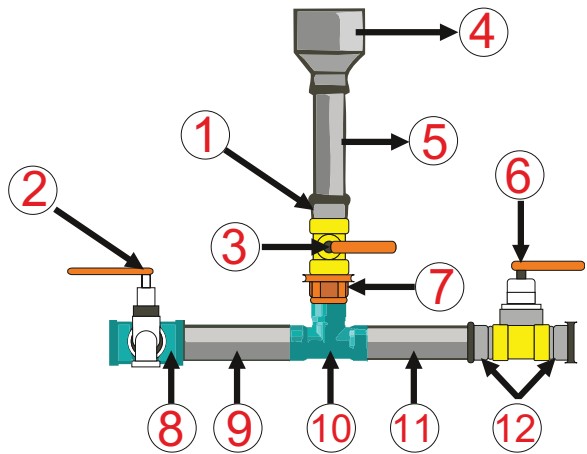

**Figure 3 Chlorinator device diagram.** (1, 12) ¾ inch male adapter threaded PVC. (2) ¼ inch garden faucet. (3, 6) ¾ inch ball register. (4) 60 × 25 inch, assembled with glue and sandpaper, and installed between the water intake port and the reservoir system. (5, 9, 11) 25 mm welded PVC tubes. (7) 25 × ¾ inch adapter. (8, 10) 25 × ¼ inch tube T-form connections.

unsatisfactory); and (2) water potability after intervention (satisfactory, unsatisfactory). For all analyses, $p < 0.05$ was considered significant.

## RESULTS

The motivation for this intervention study was a lack of water treatment offered in 65 schools located in the rural area of the city of Rio Branco, confirmed after water analysis. In this way, the intervention predicted the need for water treatment. In total, 20 schools were invited to participate in the intervention, 12 under state administration (1,986 students) and eight under municipal administration (960 students), comprising 2,946 nursery, elementary, and middle school children and staff, aged 3–60 years.

Water supply sources used by educational establishments were: 45% ($n = 9$) captured water from the Amazonas–Caçimbão well; 35% ($n = 7$) from an artesian well; 5% ($n = 1$) by water truck; and 15% ($n = 3$) from mixed sources of supply, according to water availability and climatic conditions. Water was extracted from underground and superficial wells with electric pumps.

All sources of extracted groundwater abstraction were in contravention of technical criteria of the Brazilian standards (ABNT) and without permission of public authorities. In addition, location of wells did not take into consideration potential risks of contamination, such as minimum distances from sanitary sewage systems to wells, presence of animals, residues, and land used for animal agriculture.

Only 10% ($n = 2$) of schools treated water after water collection, using 2.5% sodium hypochlorite solution, provided by the Ministry of Health (MS) and available in public health facilities in the city of Rio Branco. Another 90% ($n = 18$) did not perform any treatment. However, 60% ($n = 12$) provide industrial, filtered water troughs directly linked to the school's water reserve system.

**Table 2  Turbidity, fecal coliforms and *E. coli* results before and after environmental intervention per school.**

| Schools | Turbidity (uT)* | | Faecal coliforms | | *Escherichia coli* | |
|---|---|---|---|---|---|---|
| | Before | After | Before | After | Before | After |
| 4 | 1 | 36[1] | Present | Present | Present | Absent |
| 8 | 5 | 6[1] | Present | Absent | Present | Absent |
| 11 | 0 | 2 | Present | Present | Present | Absent |
| 16 | 1 | 15[1] | Present | Absent | Present | Absent |
| 17 | 7[1] | 2 | Present | Present | Present | Absent |
| 21 | 4 | 0 | Present | Absent | Absent | Absent |
| 29 | 273[1] | 17[1] | Present | Absent | Present | Absent |
| 34 | 16[1] | 5 | Present | Present | Present | Absent |
| 36 | 3 | 5 | Present | Present | Absent | Absent |
| 38 | 1 | 1 | Present | Absent | Present | Absent |
| 39 | 0 | 0 | Present | Absent | Present | Absent |
| 42 | 4 | 4 | Present | Absent | Absent | Absent |
| 44 | 0 | 1 | Present | Absent | Absent | Absent |
| 45 | 36[1] | 6[1] | Present | Present | Present | Absent |
| 49 | 0 | 1 | Present | Absent | Absent | Absent |
| 50 | 1 | 10[1] | Present | Absent | Present | Absent |
| 54 | 1 | 5 | Present | Absent | Present | Absent |
| 63 | 0 | 5 | Present | Absent | Present | Absent |
| 64 | 5 | 2 | Present | Absent | Absent | Absent |
| 65 | 0 | 1 | Present | Absent | Present | Absent |

Notes:
* Before: median = 1.0, 1st Qu. = 0.0, 3rd Qu. = 5.0; After: median = 4.5, 1st Qu. = 1.0, 3rd Qu. = 6.0.
[1] Values greater than the maximum permitted value (> 5 units of turbidity).

Based on inquiries directed to schools' principals, water analysis had never been conducted. In addition, schools supplied by water trucks did not require water analyses.

There was no difference in turbidity before and after the intervention ($p = 0.68$) (Table 2). Nonetheless, for total coliforms, 75% of the 20 samples analyzed were in compliance with MS Ordinance No. 2914/2011, namely absence in 100 mL at the time of treatment. There was a highly significant difference in total coliform result before and after the intervention ($p < 0.001$, Table 2). Additionally, 100% ($N = 20$) of samples were in compliance with MS Ordinance No. 2914/2011 for *E. coli*. There was a very significant difference in the *E. coli* result before and after intervention ($p < 0.001$, Table 2).

During the environmental health educational intervention process in each school, educational lectures were presented to students, teachers, and service and school support professionals. Presentations included project objectives, results of analysis of water samples collected before the intervention, the chlorine treatment process and its operation, and the instrument for measurement of free residual chlorine, in addition to information about waterborne diseases. These approaches prompted open discussions involving reflective thinking regarding personal health and hygiene issues, the role of

water quality in health and disease, sanitation of reservoirs and protection of water sources, as well as water treatment, to meet potability standards (Fig. 1).

## DISCUSSION

Basic sanitation is considered a necessary condition for economic development, for the environment and well-being of the population (*Minh & Hung, 2011*). Clean and safe water is critical to the health of all populations, regardless of location (*Whelan & Willis, 2007*).

In Brazil, the level of service of sewage networks in 2015 in urban areas of Brazilian municipalities averaged 58.0%, whereas for municipalities in northern Brazil, it was only 11.2%. For water supply networks in 2015, indices were 93.1% and 69.2%, respectively. In Acre, school attendance rate is ~60%. In rural areas, provision of water supply in households is 34.5%, 23.0% and 7.0% nationally, northern Brazil and Acre, respectively (*SNSA, 2017*).

Although Brazilian federal policy on basic sanitation requires guidelines to be followed to improve quality of life, environmental conditions and public health, and ensure provision of services to the dispersed rural population with solutions compatible with their economic and social characteristics, the proportion of people in rural areas without access to services remains high. Universalization of the basic sanitation system, as well as the National Plan for Basic Sanitation (PLANSAB), with goals to achieve 71%, 79% and 95% compliance by 2018, 2023 and 2033, respectively, are far from being realized. Instead, from 2014 to 2015, there were 31.5% reductions in investments by the state of Acre for the water supply sector (*SNSA, 2017*). Furthermore, investments for the North and Midwest regions were ~10 billion dollars until 2033, the lowest in relation to other regions of the country (*SNSA, 2013*).

The global target of the Millennium Development Goals was halving by 2015 the percentage of the population without sustainable access to safe drinking water. Although this was met by 2010 South-eastern Asia (89%), Southern Asia (90%), Western Asia (93%), Latin America and the Caribbean (95%) and Eastern Asia (96%) this goal has not been reached for the world's rural population, as the reduction from 1990 (38%) to 2015 was 16% (*United Nations, 2015*). Therefore, 17 large goals consisting of 169 smaller goals were established in Agenda 2030 for sustainable development, among them "ensure availability and sustainable management of water and sanitation for all." Of the eight smaller goals to be achieved for this goal, we highlight two: "achieving universal and equitable access to safe and affordable drinking water for all" and "support and strengthening the participation of local communities for improving water and sanitation management."

For each dollar invested in water and sanitation, four dollars in health costs are saved (*World Health Organization, 2014*). In addition to improve health and wellbeing, this intervention empowered the school community through its effective participation in water quality control. In addition, this technology represents a superior alternative to the conventional water supply system in the rural area of Rio Branco (AC), which access is often only functional in the Amazonian summer period (July/October).

In the present study, the chlorine-containing sanitizer effectively controlled pathogenic microorganisms, combating the spread of potentially transmissible diseases and ensuring a supply of potable water. This study used *E. coli* as a determinant of the microbiological quality of water in schools. Because it is a fecal coliform, this parameter is an indicator of contamination of feces of warm-blooded organisms. Its presence in the water indicates recent fecal contamination, making it unfit for consumption and contributing to the incidence of diarrheal diseases due to water-borne pathogens, including bacteria, viruses and protozoa, which are transmitted through the fecal–oral route (*Opryszko et al., 2013*). However, total coliform and turbidity parameters did not reach full compliance with standards of the Ordinance (absence in 100 mL and a maximum of 5.0 UNT, respectively). The percentage of unsatisfactory samples for total coliforms in this intervention (25%) also diverged from a study conducted in Morrinhos municipality, Brazil (*Freitas et al., 2017*), since the percentage of water samples detected with total coliforms was 49.4%.

Consumption of highly turbid water may pose a health risk (*Yasin, Ketema & Bacha, 2015*; *Hoko, 2005*), since excessive turbidity can protect pathogenic microorganisms from disinfectants, as well as allowing growth of bacteria and contributing to a significant rise of demand for chlorine (*WHO, 2011*). Components that contribute to turbidity can be related to watersheds, season (precipitation), low pressures in the distribution system and failure to routinely clean reservoirs. However, not all increases in turbidity indicate that health risks are associated with contamination (*Hsieh et al., 2015*; *Scuracchio & Filho, 2011*).

In order for chlorine action to eliminate microorganisms from the total coliform group, turbidity reduction is necessary, as it will avoid physical protection (biofilms) and transport of organic matter, which can be achieved by regular and efficient cleaning of reservoirs (*Silva, Lopes & Amaral, 2016*). Cleaning and disinfection of the water boxes was verified in a study carried out in 31 schools and day care centers in the city of São Carlos, SP, Brazil, where a reduction of 50% of total coliforms occurred after notification of these educational establishments (*Scuracchio & Filho, 2011*).

Interventions in a village in Pakistan (*Jensen et al., 2003*) and in Kitwe, Zambia (*Quick et al., 2002*) resulted in similar improvements in water quality. Another intervention in water treatment in 36 rural neighborhoods in eastern Ethiopia reduced the incidence of diarrhea among children under five in a rural population where fecal contamination had been high (*Mengistie, Berhane & Worku, 2013*).

In Brazil, there is a legal framework of the Brazilian Association of Technical Standards (ABNT) that establishes regulations regarding wells for collection of groundwater. In this study, systems of water collection were constructed in contravention of legal requirements, mainly regarding the slope of the land, the distance from the sewage system and buffering. Similarly, regulations were ignored in a study in Sichuan Province (*He et al., 2012*), as regulations in China are rarely met due to insufficient implementation and a lack of coordination between public health, education and technical departments.

The National School Feeding Program (PNAE), one of the largest school feeding programs in the world, aims to partially meet the nutritional needs of pre-school, elementary school, adult and youth education students enrolled in public and

philanthropic schools in Brazil (*Ministério da Educação, 2010*). To minimize foodborne illness in schoolchildren, each school must have a Handbook of Good Practice and Standardized Operational Procedures, developed and implemented in accordance with criteria established by RDC 216/2004 (*ANVISA, 2004*). However, there was clearly non-compliance with this technical regulation, as there was no semi-annual record of reservoir cleaning and a lack of documented water testing. These deficiencies enabled survival and multiplication of microorganisms, as well as cross-contamination of food by direct contact with water.

In Brazil, the Water Quality Surveillance Program (Vigiágua) was implemented in all states, aiming to guarantee the population access to water of a quality compatible with the drinking water standard established in the current legislation of the Unified Health System (SUS). This was intended to apply to all forms of water supply and to provide uniform data for control and surveillance in the System of Information of Surveillance of the Quality of the Water for Human Consumption (SISAGUA).

In the municipality of Rio Branco, the operation of Vigiágua is the responsibility of the Environmental Health Surveillance Division of the Municipal Health Department (SEMSA), and although the school supply form of this intervention is framed as a Collective Alternative Supply Solution of water for human consumption, the Division's responsibility was omitted in relation to this category, specifically for rural schools. Regardless, this study provided a first step toward the recognition, registration and more complete understanding of water quality in the context of the general water quality monitoring and surveillance system within the scope of the Ministry of Health, in particular the water system.

There was a lack of concern about water quality in schools, especially by students. Some teachers and principals purchased and consume bottled water. There were anecdotal reports from some students that aluminum sulfate was used as a water supply treatment measure, via the surface water body; however, there is apparently a lack of awareness regarding the concentration (ppm) that should be used. There may be increases in Alzheimer's disease or certain secondary encephalopathies of dialysis due to the consumption of water containing excessive aluminum concentrations (*Freitas, Brilhante & Almeida, 2001*; *Muniz, 2013*).

The apparent cleanliness and light color of the water were interpreted as making it safe for consumption, consistent with findings of a study in rural West Kenya (*Onyango-Ouma & Gerba, 2011*). Light-colored water as a factor associated with drinking water was related to 95% of respondents from an intervention study in Southern India (*Freeman & Clasen, 2011*).

To summarize, suggested health education programs (*Lindskog & Lindskog, 1988*) can improve students' perceptions of the importance of water quantity and quality for health, personal hygiene care and reservoirs, as well as protection of water sources.

The current intervention reinforced learning and allowed articulation between the common and scientific knowledge of the participants, as well as strengthening the partnership between the school and its users, who could act as multipliers of this

knowledge. Contrary to the study conducted in the districts of Dolakha and Ramechhap, Nepal, it was possible to observe the student's awareness of the types of waterborne diseases and modes of transmission (*Shrestha et al., 2017*). Therefore, the role of the school in transforming citizens through approaches to water-related issues, both in terms of quality and in terms of sustainable use, should be taken into account.

Several disadvantages in this intervention should be recognized and taken into account while interpreting results of this study. Administration of 65% calcium hypochlorite in the simplified chlorinator was dependent on the availability of a school professional; in their absence, its replacement does not occur. Other deficiencies included a lack of regular cleaning of reservoirs, location of rural schools in an area not accessible in winter due to lack of asphalt paving, thereby preventing regular monitoring. However, there were also some advantages: the technology cost less than US$30, was easily established and operated, and was a lower-cost option for schools that need to buy water from water trucks or other similar sources.

We recommend that school management plans include actions involving approaches to improve education in health, safety, and water quality in order to sensitize schoolchildren to water as potential source despite looking apparently clear. Likewise, public policies should ensure expansion of interventions on water treatment to other educational establishments located in rural areas, as well as systematic monitoring of water quality by municipal authorities. Furthermore, reservoir sanitation should also be done regularly. Finally, a Manual of Good Practices and Standard Operating Procedures should be prepared, aiming at the nutritional protection of school children.

Future assessments may provide important data on the sustainability of this intervention and efforts to provide safe drinking water. In this way, it is recommended to conduct complementary studies to identify pathogenesis of strains of *E. coli*, emphasizing the academic and scientific importance of this type of analysis for adoption of public health policies.

## CONCLUSION

The actions carried out in this intervention have considerably improved the water quality of the schools, thus reversing the health vulnerability due to inadequate water provided to the school community in the rural area. Comprehensive activities, including training, educational lectures, installation of equipment, supply of materials and supplies (65% calcium hypochlorite, reagents, etc.) were fundamental to success. This intervention supported the human right of access to water, and it contributed to the health, well-being and food and nutritional security of the school community involved.

Since health education is one of the main actions for health promotion, we recommend that: (1) the Health Program in the school consider the theme of water as a potential source of disease transmission, promoting dissemination of this information; (2) other educational institutions adopt this model of water treatment, since actions of the

School Health Program, established in Interministerial Ordinance nº 1,565 of April 25, 2017, does not contemplate this theme; and (3) government authorities acquire the necessary inputs for treatment of water and distribute them to schools, as well as perform regular monitoring of water quality by the Environmental Surveillance of the Municipal Health Department.

## ACKNOWLEDGEMENTS

To Herman Barkema and John Kastelic for generously helping with improvements in the manuscript text.

### Funding

This work was supported by the Acre Project—Health in the Western Amazonia (multi-institutional agreement process n. 007/2015 SESACRE—UFAC—FMABC), the São Paulo Research Foundation (FAPESP; n. 2014/09774-1), and the Health National Foundation—Rio Branco Government (agreement n. 795710/2013). There was no additional external funding received for this study. The funders had no role in study design, data collection and analysis, decision to publish, or preparation of the manuscript.

### Grant Disclosures

The following grant information was disclosed by the authors:
Acre Project—Health in the Western Amazonia: n. 007/2015 SESACRE—UFAC—FMABC.
São Paulo Research Foundation: n. 2014/09774-1.
Health National Foundation—Rio Branco Government: n. 795710/2013.

### Competing Interests

The authors declare that they have no competing interests.

### Author Contributions

- Maura Regina Ribeiro conceived and designed the experiments, performed the experiments, contributed reagents/materials/analysis tools, prepared figures and/or tables, authored or reviewed drafts of the paper, approved the final draft.
- Luiz Carlos de Abreu conceived and designed the experiments, authored or reviewed drafts of the paper.
- Gabriel Zorello Laporta conceived and designed the experiments, analyzed the data, prepared figures and/or tables, authored or reviewed drafts of the paper, approved the final draft.

### Data Availability

  The laboratory reports with results of water potability before and after intervention are provided in a Supplemental File.

## Supplemental Information

Supplemental information for this article can be found online at http://dx.doi.org/10.7717/peerj.4993#supplemental-information.

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
