# Peer review of "Drinking water and rural schools in the Western Amazon: an environmental intervention study"

_PeerJ, doi:10.7717/peerj.4993_

## Round 0.1 · original submission · Minor Revisions

Authors are required to resubmit the paper after incorporating comments by Reviewer 2 and 3.

·

Basic reporting

The paper titled Drinking water and rural schools in the Western Amazon: an environmental intervention study is an interesting research work.
The authors have clearly highlighted the problem of study area.

Experimental design

The submission has clearly defined the research question, which is very much relevant and meaningful. The investigation have been conducted rigorously and to a high technical standard. The research have been conducted in conformity with the prevailing standards in the study area.

Validity of the findings

The conclusions have been appropriately stated and have been connected to the original question investigated, and are result oriented

Additional comments

This is a meaningful research paper with a great potential to be published in this journal.

Reviewer 2 ·

Basic reporting

The paper is based on the study of a sanitary intervention in rural areas of Brazilian western Amazon. The main outcome of the study can be considered as not really significant or without a clear perspective of an impact since it dedicates too much effort just in order to prove a predictable reduction of coliforms in water supply after the implementation of a simplified chlorinator. Besides this particular hypothesis the paper refers to such intervention as a social technology, but instead of treating such issue as an inter/multidisciplinary concern the paper does not explore the challenge of implementation which could bring a more contextualized and rich study considering the peculiar conditions and socio-environmental constraints in the rural settlements. Therefore, dealing with the complexity of implementation of a social technology would be a task for a different study design.

Experimental design

Insufficient to deal with sanitation as a social technology.

Validity of the findings

Findings are not relevant.

Additional comments

The simple confirmation of coliform reduction can be considered as a matter of how sanitation and health concerns have been dealt as a surpassed paradigm, but of course, it is not something really solved, indeed there is a need for interdisciplinary approaches to bring new outcomes and solutions.

Reviewer 3 ·

Basic reporting

The document has a clear intervention study and its evaluation and perspectives.
The review presented is well fitted to the subject supporting the proposal and in the identification of comparative papers to evaluate the results.
About the structure:
• An adequate abstract presents background, methods, results and discussion.
• The methods are well described with:
o Study design, studied area and population, intervention process, water parameters and data analysis.
• The results are clear and
• The discussion relates the results obtained with other studies and important references.
• The conclusion presents the findings of the intervention study, could include the future directions.
The figures are well labelled and described:
• In Figure 1 - Infographic of the environmental intervention study, it would be interesting to indicate the initial data collection as a step in the environmental intervention study.
• Figure 2 – Indicates the area of study, it would be interesting to present some spatial reference (coordinates, name of location or limits).
• Figures 3 to 5 present a synthesis of the Results obtained before and after intervention.
The Raw data supplied is consistent with the paper presented.

Experimental design

The proposed research scope is well defined as an intervention investigation. The method is clear and pertinent, with an emphasis on the application and its replicability.

Validity of the findings

The data included corroborate the findings demonstrating the improve in the school’s water quality using the intervention proposed. The literature arguments are consistent. The descriptive statistics is clear, and the graphics contribute in the presentation.

Additional comments

Suggestions:
The English language could be improved to make the text more precise. Some examples are indicated.
• Line 24 “the water quality of the schools” - schools water quality
• Line 32 “about three out of ten people (2.1 billion)” - worldly about three out of ten people (2.1 billion)
• Line 40 “As the rural population remains living in adverse conditions and deprived of drinking water,” - As most rural population remains living in adverse conditions and deprived of drinking water

Figure 1 - Infographic of the environmental intervention study, it would be interesting to indicate the initial data collection as a step in the environmental intervention study.

Figure 2 – Indicates the area of study, it would be interesting to present some spatial reference (coordinates, name of location or limits).

---

## Round 0.2 · Minor Revisions

As you may notice, I have taken this manuscript over from another editor who is currently unavailable. I like the study and the manuscript, but I have two comments/requests.

1) The data collected on the 20 schools before and after the intervention are paired by school. The statistical analysis therefore also needs to be paired (a Chi-square or Fisher's Exact test ignore paired data). Therefore please conduct a McNemar's test.

2) The results presented in Figures 3-5 can better be presented together in one Table. I would suggest not only present the % above the cut-offs, but also mean (or median if the data are not normally distributed) and a 95% confidence interval before and after the intervention.

Please feel free to consult with me on the exact design of the table.

Reviewer 3 ·

Basic reporting

The suggestions proposed were considered and the improvements realized. The document is clear and unambiguous. The references are presented and the research fits into the field of knowledge. The article structure is properly organized. Raw data is available and clearly presented. Figures and table are relevant and contribute to the understanding of the paper.

Experimental design

The paper presents a research project with environmental intervention for the treatment of water for human consumption, in public schools.
The methods are presented, the proposal is consistent, and the meaning of the water quality control and measures are significant. Results obtained support the analysis and the procedures are understandable and could be replicated.

Validity of the findings

The data presented is clear and the related analysis consistent.
The presentation made contribute to the literature by showing the environmental intervention for the treatment of water for human consumption and the relation to literature contributing to the set of studies applied in the area.
.

Additional comments

I suggest that you improve the writing at lines 41- 47, to make it clearer.

---

## Round 0.3 · Minor Revisions

Thank you for adding the results of the statistical analyses and putting the data of the figures in the tables 2 and. However, these tables can be combined in one table. Please do so.

Additionally, the English of your manuscript really needs to be edited. I would be happy to do so (with the assistance of an Anglophone colleague who is a very experienced editor). I will need the manuscript as a Word file to do so. Please e-mail it to [email protected] and we'll do this in the next couple of days. See it as a "pay-back" for the 3 great Brazilian graduate students I have....

---

## Round 0.4 · Minor Revisions

I'm just sending you a 'Minor Revisions' because you indicated that you unintentionally had submitted the manuscript while we are editing it.

---

## Round 0.5 · accepted · Accept

It was my pleasure helping you with the revisions.